

# Vertical extrapolation of ASCAT ocean surface winds using machine learning techniques

Daniel Hatfield[1], Charlotte Bay Hasager[1], and Ioanna Karagali[2]

[1]Department of Wind and Energy Systems, Technical University of Denmark, Frederiksborgvej 399, 4000 Roskilde, Denmark
[2]Danish Meteorological Institute, Lyngbyvej 100, 2100 København Ø, Denmark

**Correspondence:** Daniel Hatfield (dhat@dtu.dk)

**Abstract.** The increasing demand for wind energy offshore requires more hub-height relevant wind information while larger wind turbine sizes require measurements at greater heights. In situ measurements are harder to acquire at higher atmospheric levels; meanwhile the emergence of machine-learning applications has led to several studies demonstrating the improvement in accuracy for vertical wind extrapolation over conventional power-law and logarithmic profile methods. Satellite wind retrievals supply multiple daily wind observations offshore, however only at 10 m height. The goal of this study is to develop and validate novel machine-learning methods using satellite wind observations and near-surface atmospheric measurements to extrapolate wind speeds to higher heights. A machine-learning model is trained on 12 years of collocated offshore wind measurements from a meteorological mast (FINO3) and space-bourne wind observations from the Advanced Scatterometer (ASCAT). The model is extended vertically to predict the FINO3 vertical wind profile. Horizontally, it is validated against the NORA3 meso-scale model reanalysis data. In both cases the model slightly over-predicts the wind speed with differences of 0.25 and 0.40 m s$^{-1}$ respectively. An important feature in the model training process is the air-sea temperature difference, thus satellite sea surface temperature observations were included in the horizontal extension of the model, resulting in 0.20 m s$^{-1}$ differences with NORA3. A limiting factor when training machine-learning models with satellite observations is the small finite number of daily samples at discrete times; this can skew the training process to higher/lower wind speed predictions depending on the average wind speed at the satellite observational times. Nonetheless, results shown in this study demonstrate the applicability of using machine learning techniques to extrapolate long-term satellite wind observations when enough samples are available.

## 1 Introduction

Wind observations at heights relevant for the operation of modern offshore wind farms, i.e. 100 m and more, are required to optimize their positioning and layout. Direct measurements offshore, especially in deep water locations, are costly and thus, only available for limited time periods. Traditionally, meteorological masts (met. masts) are used to characterize the ambient



wind speeds, however with the increasing size of wind turbines and water depths these become more expensive to install (MacAskill and Mitchell, 2013).

Wind lidars can measure the line of sight wind speed at distances from a few centimeters to several kilometers on land,
floating buoys or ferries at sea or in orbit on satellites (Clifton et al., 2018). Floating lidar systems can act as a substitute to met masts, as they are able to measure wind profiles from near the ocean surface and up to 275 m with high sampling frequency. However many of the existing floating lidar system datasets are privately owned or of shorter time periods not suitable to characterize the inner-annual wind variations (Gottschall et al., 2017).

Numerical models provide wind simulations over long time periods and at many levels in an area of interest. For wind
energy applications, such simulations do not always accurately reproduce the actual wind variability. Additionally, the errors associated with simulated winds from numerical models are not accurately characterised, mainly due to the sparsity of offshore wind data (Hahmann et al., 2015). This adds uncertainty to wind resource mapping with larger errors found at more complex offshore sites (Peña et al., 2011).

Satellite wind retrievals provide observations of the wind field over large spatial domains and extensive time periods yet their
temporal resolution, e.g. up to a few times per day at best, is limited compared to model simulations and in situ measurements. Synthetic Aperture Radar (SAR) and scatterometer wind measurements have been used to characterize offshore wind resources (Karagali et al., 2018a; Remmers et al., 2019; Hasager et al., 2020; Ahsbahs et al., 2020). ASCAT scatterometer winds were compared to numerical model simulations(Karagali et al., 2018b) and ferry lidar measurements showing better agreement than the meso-scale model simulations (Hatfield et al., 2022). ASCAT winds are optimized for consistent wind measurement
accuracy (Verhoef et al., 2017), stability (Rivas et al., 2017) and bias (Belmonte Rivas and Stoffelen, 2019). Nevertheless, satellite wind observations are representative at the 10m height which is not directly applicable for wind energy purposes at hub heights. Badger et al. (2016); Hasager et al. (2020) extrapolated surface winds to higher atmospheric levels over the European seas using the long-term stability correction from Kelly and Gryning (2010); results were promising when compared to in situ wind measurements offshore.

Machine learning is a novel method for predict wind speeds at different heights from in situ measurements onshore (Türkan et al., 2016; Mohandes and Rehman, 2018; Vassallo et al., 2020; Bodini and Optis, 2020) and offshore (Vassallo et al., 2020; Optis et al., 2021). Türkan et al. (2016) compared seven different machine-learning algorithms predicting 30 m wind speeds from 10 m wind speed data with Root-Mean-Square-Errors (RMSE) ranging from 0.2 m s$^{-1}$ to 0.9 m s$^{-1}$, reporting the RandomForest and Multilayer Perceptron as the best performing ones. Mohandes and Rehman (2018) used a Deep Neutral
Network to extrapolate wind lidar data, providing better estimates than classical non machine learning methods with improvement on the power-law predictions of up to 15% at 100 m heights. Vassallo et al. (2020) used an Artificial Neural Network to extrapolate wind speeds over a variety of terrains, improving accuracy by up to 65% and 53% compared to the logarithmic profile and power law methods, respectively. The machine-learning approach was used by Optis et al. (2021) to extrapolate offshore floating lidar wind speed measurements, demonstrating improved performance compared to Weather Weather Research
& Forecasting Model (WRF) model data, logarithmic profile methods, single column model data and the extrapolation method





of Badger et al. (2016). de Montera et al. (2022) used machine learning techniques to improve bias on SAR wind retrievals and to extrapolate the resultant SAR winds to hub heights to obtain wind power maps around the training area.

Although results from Mohandes and Rehman (2018); Vassallo et al. (2020) showed better performance of the machine learning models compared to the conventional methods of profile extrapolation, these studies were assessed at the sites where the model training took place. A "round-robin" approach to properly validate the machine-learning based vertical extrapolation was suggested by Bodini and Optis (2020); this involves training the model at the given site of interest and assessing it at other sites, some distance away from the original location. Bodini and Optis (2020) reported an increase in Mean Absolute Error (MAE) by 10%-15% at distances of 50-100 km, stating that the machine-learning based approach outperformed the classical extrapolation methods in all atmospheric stability conditions.

The aim of this study is to assess the potential of using machine learning models with two-dimensional wind field observations at lower atmospheric levels in order to predict the wind at great heights. More specifically, ASCAT ocean surface wind retrievals are extrapolated using a machine-learning model to higher atmospheric levels, directly relevant for wind energy applications. Sensitivity analyses on the input data used for training the model are performed and special attention is given to the impact of input data sampling frequency to the training model performance. Finally, following the "round-robin" approach, this study also aims at spatially assessing the performance of the machine learning methods, i.e. to a nearby met mast and around an area surrounding the training site.

Section 2 describes the data sets, study area and machine-learning model. Section 3 describes the model training process at three sites and with prediction of mean wind profiles at one site, including outcomes of the round-robin approach for validation. Discussions on the findings and conclusions are available in sections 4 and 5, respectively.





## 2   Data and Methods

### 2.1   ASCAT

The Advanced Scatterometer (ASCAT) is an instrument on the Meteorological Operational (MetOp) satellites, operated by the European Organization for the Exploitation of Meteorological Satellites (EUMETSAT) (Verhoef and Stoffelen, 2019). ASCAT was launched subsequently on Metop-A in October 2006, Metop-B in September 2012 and Metop-C in November 2018. ASCAT is a real aperture radar operated in the C-band (5.255 GHz) consisting of two sets of three vertically polarised antennas separated by $45°$. These beams measure a 550-km swath with a 700-km nadir gap, where each swath is divided into 41 Wind Vector Cells (WVCs) covering a 12.5-km grid of the sea surface. As backscatter increases with increasing sea surface roughness (Stoffelen, 1996), in each WVC the backscattered power from the observed area is used to estimate the normalized radar cross section (NRCS, $\sigma_0$) (Martin, 2014). The NRCS is the relation between the received and transmitted power which is dependent on the radar settings, the atmospheric attenuation and the ocean surface characteristics (Chelton et al., 2001). A geophysical model function (GMF), i.e. an empirically derived function based on the local measurement geometry, relates the mean wind vector in a WVC to the NRCS (Stoffelen et al., 2017; de Kloe et al., 2017; Vogelzang et al., 2017).

ASCAT products include wind speed and direction at 10 m above the sea surface and for the purpose of the present study the Near-Real-Time (NRT) 12.5-km wind product (from 2010–2015 WIND_GLO_WIND_L3_REP_OBSERVATIONS_012_005 from 2007–2015 and WIND_GLO_WIND_L3_NRT_OBSERVATIONS_012_002 from 2016 onwards) was used from January 1, 2010 to December 31, 2021. This 12.5 km product has a standard deviation of 1.7 m s$^{-1}$ and a bias of 0.02 m s$^{-1}$ (Verhoef and Stoffelen, 2019). Data are produced by the Royal Netherlands Meteorological Institute (KNMI) for and are distributed by the Copernicus Marine Service (https://marine.copernicus.eu). For the area of interest, ASCAT provides a measurement 94% of the total time period. There are from 1 to 5 observations daily with a higher frequency of observations in the latter half of the time period due to the coverage of all three MetOp satellites although MetOp-A was decommissioned on November 30, 2021.

### 2.2   FINO meteorological masts

The German Forschungsplattformen In Nordund Ostsee (FINO) project began in the early 2000s (fino.bsh.de), with the installation of offshore met masts in the North and Baltic Seas to study the wind climate over long time scales (Leiding et al., 2016). Meteorological parameters are recorded at frequencies of 1-10 Hz, and averaged in intervals of 10-30 min. Observations were used during the period 1 January 2010 to 31 December 2021. Details on the masts are available in Table 1 and are shown in Figure 1.

FINO1 is situated in the North Sea approximately 45 km to the north of Borkum, Germany and in the immediate vicinity of the wind farms Alpha Ventus and Borkum Riffgrund. The average wind speed is 9.9  m s$^{-1}$ from 2010 to 2021 at 91 m with a south-westerly prevailing wind. All measurements are available over 90% of the period of interest, except the water temperature (WT) with 84% availability.





**Table 1.** Characteristics of FINO masts with the heights of available measurements for various meteorological and oceanographic parameters.

|  | FINO1 | FINO2 | FINO3 |
|---|---|---|---|
| Latitude° | 54.01 | 55.01 | 55.20 |
| Longitude° | 06.58 | 13.15 | 07.15 |
| Bathymetry (m) | 30 | 35 | 25 |
| Wind Speed (m) | 34, 41:10:91 | 32:10:102 | 31:10:101, 107 |
| Wind Dir (m) | 34, 50, 70, 90 | 31, 51, 71, 91 | 29, 101 |
| Air Pressure (m) | 21 | 30 | 23 |
| Air Temperature (m) | 34 | 30 | 29 |
| Relative Humidity (m) | 34 | 30 | 29 |
| Sea Surface Temperature (m) | -2 | -2 | -2 |

FINO2 is located in the Baltic Sea, within 3 km north of the EnBW Baltic 2 wind farm and 33 km north of the Rügen island. The average wind speed is 9.6 m s$^{-1}$ at 102 m with a south-westerly prevailing wind. All relevant measured variables are available 90% of the 12-year period of interest with the exception of WT with 64% data availability.

FINO3 is located in the North Sea to the west of the DanTysk wind farm, 70km from the island of Sylt. The average wind speed is 9.6 m s$^{-1}$ at 107 m over the entire measurement period with a westerly prevailing wind. All measured quantities show a data availability of 85% except WT (76%).

## 2.3 Satellite Sea Surface Temperature

Besides the water temperature measurements at the met mast locations, which are typically taken at some depth below the
surface and are representative of that specific location, space-borne infrared radiometers provide extensive spatial and temporal coverage of the actual sea surface temperature, i.e. SST$_{skin}$, which is typically converted to SST$_{sub-skin}$ and is considered representative of the few top millimeters of the water surface (Donlon et al., 2007). The Copernicus Marine Environment Monitoring Service (CMEMS) releases a suite of level 4, gap-free products with regional and global coverage, representative of the SST foundation temperature, i.e. the temperature free of diurnal warming or nocturnal cooling, typically at the base of
the sub-skin layer (Donlon et al., 2007). For the purposes of the present study, the Baltic Sea/North Sea SST (DMI level 4 (L4) SST) reprocessed L4 analysis was used; it is a gap-free satellite foundation SST analysis created by the Danish Meteorological Institute (DMI) Optimal Interpolation (OI) system (Høyer and She, 2007). The product is available from 1st January 1982 to 31st May 2021 - it is being temporally extended at regular intervals - on a regular grid with 0.02° resolution. It provides an estimate of the foundation SST with uncertainty estimates, which is the SST free of diurnal variability (Høyer and Karagali,
2016). See CMEMS (2022) for further details.

Data are produced by the Danish Meteorological Institute (DMI) for and are distributed by the Copernicus Marine Service (product ID SST-BAL-SST-L4-REP-OBSERVATIONS-010-016, https://resources.marine.copernicus.eu/). To diversify from



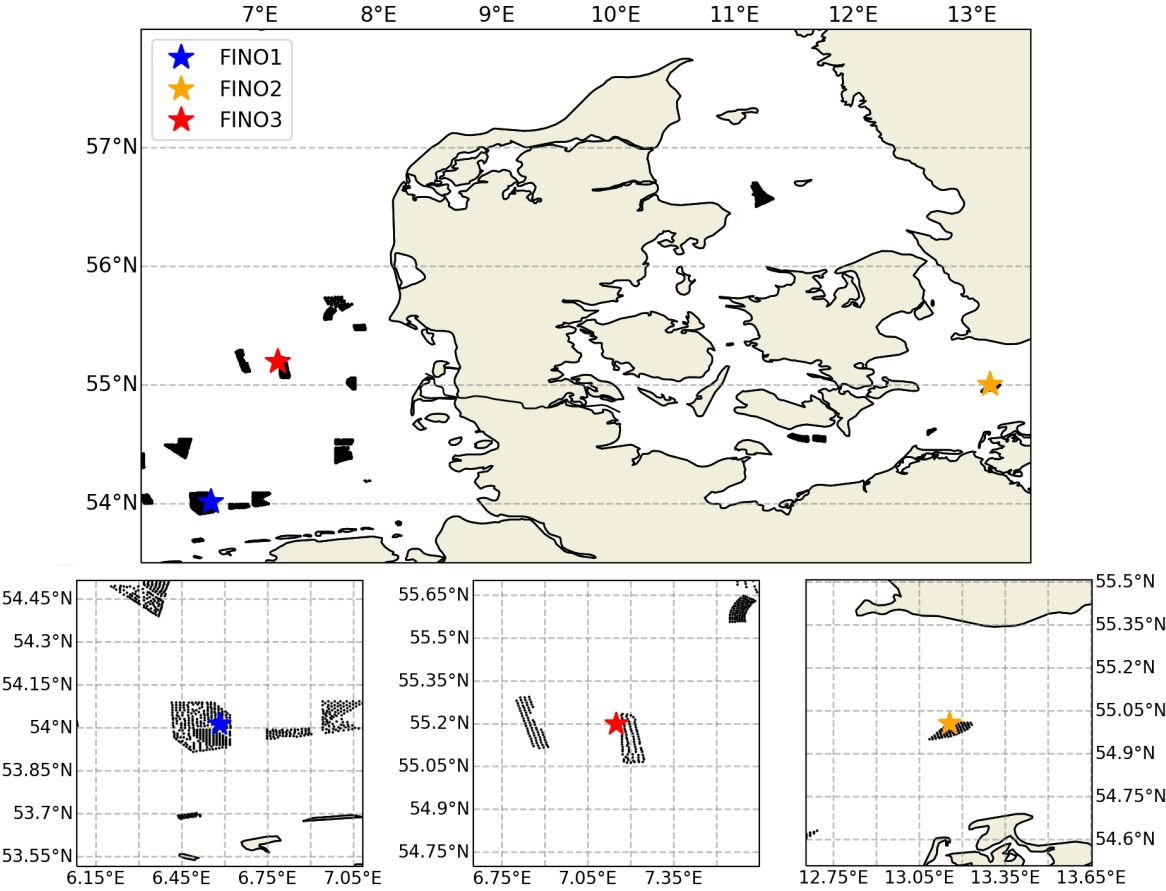

**Figure 1.** Map of the study area with the FINO mast locations in the North and Baltic Seas (top). The black rectangles represent nearby offshore wind farms. The bottom panels show close-ups for the met mast locations with black dots representing individual wind turbines.

the water temperature measurements available at each meteorological mast site, this product will be referred to as DMI L4 SST for the remaining of this manuscript. For spatial matching with ASCAT and since the spatial resolution of the DMI L4 SST product is 0.02°, a 3x3 grid of SST observations centered in the ASCAT WVC were averaged for each WVC and re-mapped to the ASCAT coordinates.

## 2.4 Simulated Wind Datasets

The NORwegian hindcast Archive (NORA3) is a re-analysis hindcast dataset with a 3-km spatial resolution, available from 1984 to 2021 for the Norwegian, the North and the Barents Sea. NORA3 is dynamically downscaled from the European Centre for Medium-Range Weather Forecasts (ECMWF) ERA5 reanalysis (Hersbach et al., 2020), using the Numerical Weather Prediction (NWP) model HIRLAM–ALADIN Research on Mesoscale Operational NWP in Euromed—Applications of Research



to Operations at Mesoscale (HARMONIE-AROME). Three nested domains were used (18 km, 6 km and 2 km horizontal resolution), with a model-integration time of 4 years (2004-2007), and a temporal resolution of 1 hour.

The New European Wind Atlas (NEWA) dataset, like NORA3, has a 3-km spatial resolution and is derived from ERA5 reanalysis (Hersbach et al., 2020), however it is down-scaled using the WRF model with no data assimilation (Hahmann et al., 2020; Dörenkämper et al., 2020).

For the purpose of this work, only the year 2018 was considered for comparisons due to the concurrent availability of ASCAT and the New European Wind Atlas (NEWA) dataset (Witha et al., 2019). Due to the different spatial resolution of NORA3 and ASCAT, NORA3 was re-sampled according to the ASCAT grid; for a given ASCAT WVC, a 3x3 grid of NORA3 grid points centered around that ASCAT WVC was averaged and remapped to the ASCAT coordinates.

## 2.5 Random Forest Model

A simple ensemble-based regression tree method known as a random forest model (Breiman, 2001; Hastie et al., 2009) was used in the present study for wind speed extrapolation. A random forest is a collection of decision trees which are trained on random subsets of a training dataset. From the input data, the algorithm generates a forest of $N$ trees $\{T_1(X), T_2(X), ..., T_N(X)\}$ using a $k$ dimensional vector input $X = \{x_1, x_2, ..., x_k\}$ and a target dataset $Y = \{y_1, y_2, ..., y_k\}$. These $N$ independent trees predict a final value which is then averaged across all trees: $\overline{y} = \frac{1}{N} \sum_{n=1}^{N} T_n(x)$ where $x$ is a sample in the testing set and $\overline{y}$ is the final value. The `RandomForestRegressor` module in Python's scikit-learn package (Pedregosa et al., 2011), previously used for wind extrapolation in Bodini and Optis (2020); Optis et al. (2021), was implemented for this study.

Water and air temperature, relative humidity and air pressure measurements, averaged every 30 minutes, from each of the three FINO met masts were used as input data for the model training along with instantaneous wind speed, cosine of wind direction, time of day and month from ASCAT (see Table 2). The associated number of concurrent samples, i.e. match-ups, are shown in Table 3. The fewer samples for the FINO2 mast are associated to the later starting date of WT measurements (2013), resulting in a shorter training period compared to the other two masts, i.e. 7 years for FINO2, 11 years for FINO3 and 14 years for FINO1.

While model parameters are "learned" during the training phase, *hyper-parameters* are set before the training to create a more accurate algorithm. Hyper-parameter tuning relies on experimental results of combinations of model parameters to evaluate the performance of each model. To avoid over-fitting the model, the K-Fold cross-validation method is applied. The data is split into testing and training sets which is split further into five subsets, *K*. The model is trained iteratively K times, evaluating on the K-th fold, changing on each iteration. The hyper-parameters allowed to vary and their associated ranges are outlined in Table. 4. This procedure is repeated for each of the FINO masts.



**Table 2.** Input features and heights used to train the random forest model at the FINO met masts.

| Source | Input feature | Acronym | FINO1 heights [m] | FINO2 heights [m] | FINO3 heights [m] |
|---|---|---|---|---|---|
| FINO | Air Pressure [hPa] | AP | 21 | 30 | 23 |
| | Air Temperature [°C] | AT | 34 | 30 | 29 |
| | Relative Humidity [%] | RH | 34 | 30 | 29 |
| FINO | Water Temperature [°C] | WT | 0.5 | 0.5 | 0.5 |
| | Air-Sea temperature difference [°C] | AT - SST (WT) | - | - | - |
| DMI L4 SST | Sea Surface Temperature [°C] | SST | 0 | 0 | 0 |
| ASCAT | Wind Speed [ m s$^{-1}$] | WS | 10 | 10 | 10 |
| | Cosine of Wind Direction [°] | WD | 10 | 10 | 10 |
| | Time of day (hour) | H | - | - | - |
| | Month | M | - | - | - |

**Table 3.** Total number of samples used in the random forest model training from each FINO mast.

| | Total data | Concurrent data with ASCAT | Data used in model training | Data used for validation | Period of data availability |
|---|---|---|---|---|---|
| FINO1 | 157129 | 6177 | 4942 | 1235 | 2007-01-01 to 2021-07-31 |
| FINO2 | 121774 | 4618 | 3694 | 924 | 2013-04-17 to 2020-11-06 |
| FINO3 | 137577 | 5739 | 4592 | 1147 | 2010-01-22 to 2021-07-31 |

**Table 4.** Hyperparameter input range for model cross-validation.

| Hyperparameter | Value range |
|---|---|
| Number of estimators | 50-1000 |
| Minimum number of samples per split | 2-10 |
| Minimum number of samples per leaf | 1-10 |
| Maximum number of features per tree | 1-9 |
| Maximum depth | 5-30 |



**Table 5.** Metrics of the random forest model (ML) and NEWA WRF dataset, compared to the wind measurement at the height nearest to 100 m at each of the FINO met masts. Random forest models were trained at each of the FINO met masts using the lowest atmospheric variable measurements available at each height. Results in bold represent the best results for each evaluation metric.

|  | Height [m] | N | $R^2$ | RMSE [ m s$^{-1}$] | MAE [ m s$^{-1}$] | bias |
|---|---|---|---|---|---|---|
| NEWA at FINO1 | 100 | 49830 | 0.65 | 2.39 | 1.78 | 0.075 |
| ML at FINO1 | 91 | 6180 | 0.82 | 1.90 | 1.40 | 0.010 |
| NEWA at FINO2 | 100 | 50013 | 0.65 | 2.32 | 1.76 | 0.040 |
| ML at FINO2 | 102 | 4618 | 0.78 | 1.83 | 1.37 | -0.009 |
| NEWA at FINO3 | 100 | 51463 | 0.77 | 1.93 | 1.41 | **0.003** |
| ML at FINO3 | 101 | 5739 | **0.93** | **1.23** | **0.90** | -0.004 |

## 3 Results

### 3.1 Site selection for random forest model training

The random forest model was parameterized and trained at each of the three FINO sites, in the North and Baltic Seas. Table 5 shows the metrics of the predicted wind speeds at the highest available heights of each mast: 91 m at FINO1, 102 m at

FINO2 and 107 m at FINO3. The models trained at FINO1 and FINO2 have an RMSE ~1.8 m s$^{-1}$ whereas for FINO3 the RMSE is lower, i.e. ~1.2 m s$^{-1}$. The model trained at FINO3 also has the lowest Mean Absolute Error (MAE) as well as the highest coefficient of determination (0.93). At all sites, biases were negligible with the lowest value from the machine-learning output of =0.004 at FINO3. Not that the biases were calculated with respect to the met masts i.e., $(\overline{U}_{pred} - \overline{U}_{mast})/\overline{U}_{mast}$ The NEWA dataset also has lowest RMSE, MAE, bias as well as the highest coefficient of determination at the FINO3 site

compared to the other two. The machine-learning model shows lower RMSE and MAE compared to the NEWA dataset at all FINO sites.

 Feature importance for the random forest model is calculated based on the increase or decrease in error when permuting over the value of a particular feature. If permuting the values causes a large change in the mean square error (MSE), the feature is an important training criterion for the model. The left panel of Figure 2 shows the contributions of various input features to

the mean model accuracy, with a decrease over the training period for the FINO3 dataset. As expected, the ASCAT 10 m wind speed is the most important feature while contributions from the other input variables are small to negligible. This behaviour is consistent for the training process at all sites, with the air-sea temperature difference consistently being the second most important training feature. Nonetheless, including the air-sea temperature difference as a feature reduces the overall RMSE by around 20% at all sites. The right panel of Figure 2 shows statistics of the predictions at a height of 107 m. Training the dataset

at lower heights results in an overall lower RMSE and a higher contribution from the lower atmospheric variables in terms of feature importance, i.e. the air pressure shows higher contribution to the training for heights up to 80 m (not shown).

 In summary, the model training procedure repeated at the three FINO sites showed best statistics at FINO3 (Table 5); there, less wind farms exist in the vicinity of the meteorological mast compared to the other two sites (Figure 1) and the highest data





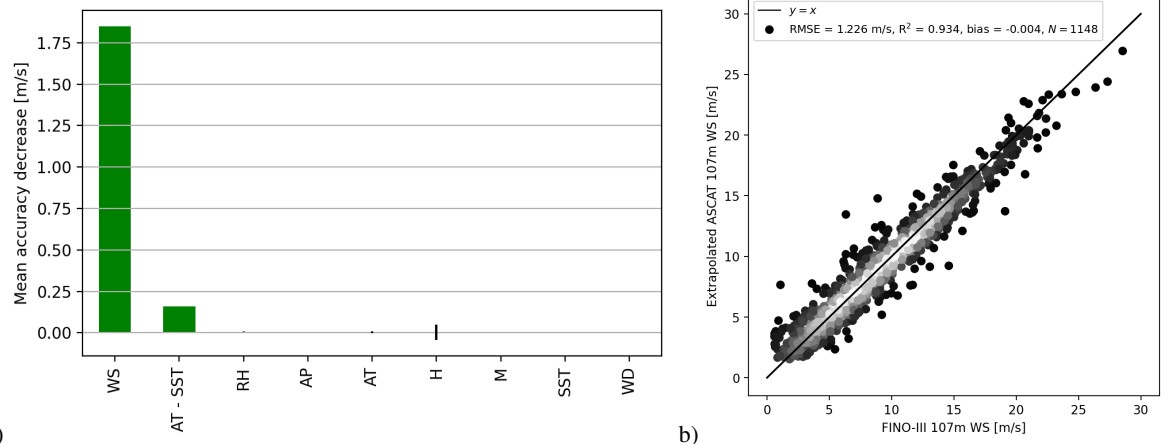

**Figure 2.** a) Mean accuracy decrease of the mean square error contribution from the training variables. The small vertical black bars represent the standard deviation of the mean training dataset. b) Scatter plot of the predicted ASCAT 107m wind speed (y-axis) versus FINO3 107m wind speed measurements (x-axis), based on the 20% validation dataset not used in the model training process.

availability of wind speed measurements is recorded. For these reasons, focus is given only on this site for the remainder of this study.

### 3.2 Wind profiles reconstruction

The random-forest model (RFM) was used to reproduce the mean wind profile at FINO3, shown in the left panel of Figure 3, along with that derived from measurements on site. The observed wind profile (red dots) shows very low shear, increasing from 8.7 to 9.7 m s$^{-1}$ between 31 m and 107 m. The RFM (black line) performs very well at predicting the mean wind profile. The right panel of Figure 3 shows the mean wind speed residuals, i.e. the difference between the RFM wind profile minus the observed one, at each height. At lower heights, from 31 m to 51 m, the model reproduces the wind speeds with a slight over-estimation of just over 0.03 m s$^{-1}$ while residuals marginally increase at higher heights indicating a slight over-estimation of the wind profile derived from the RFM. Overall the RFM could reproduce the collocated wind profile at FINO3 with overall very low residuals and slight deviations at higher heights.

### 3.3 Round robin approach at FINO1 & FINO3

The round robin approach used here aimed at applying the RFM trained at FINO3 to estimate the mean wind speed at FINO1, located 136 km away. Moreover, comparisons with the measurements at FINO1 were performed. For validation purposes, the RFM was optimized through training at the 91 m height of FINO3 using the satellite-based DMI L4 SST product (see 2.3), instead of the water temperature (WT) measured on site. This optimized RFM was extended to the location of FINO1, where only the ASCAT wind speed/direction and the DMI L4 SST were substituted for the FINO1 site; all other model features, i.e. air temperature, pressure and relative humidity, were assumed to be static, retaining the values used at FINO3.





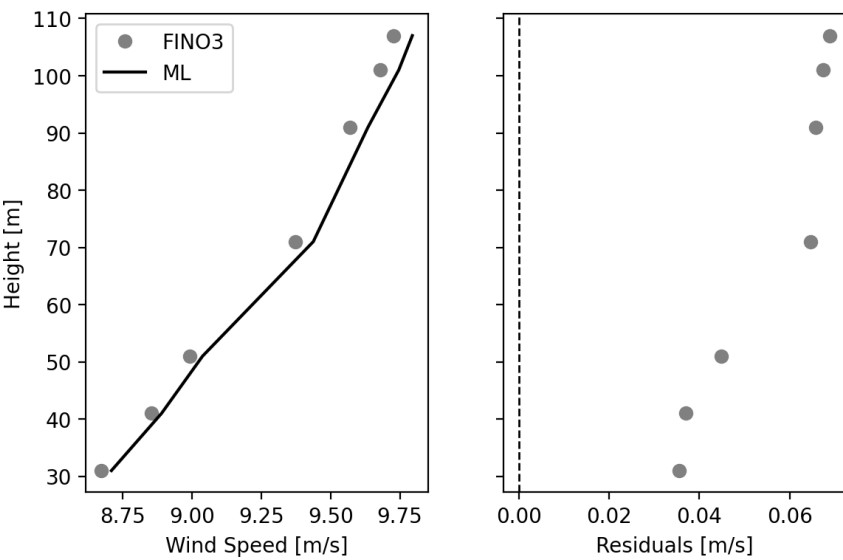

**Figure 3.** 2010-2021 mean wind profile at FINO3 from the RFM (line) and the corresponding measurements as dots (left). Wind speed difference between the RFM and the observations (right).

**Table 6.** Round robin model evaluation from FINO3 and FINO1 using the DMI L4 SST product and water temperature (WT) measurements at each site.

| | Including DMI L4 SST | | | | Including water temperature at mast | | | |
|---|---|---|---|---|---|---|---|---|
| | FINO1 | FINO3 | FINO3 to FINO1 | FINO1 to FINO3 | FINO1 | FINO3 | FINO3 to FINO1 | FINO1 to FINO3 |
| RMSE [ m s$^{-1}$] | 1.803 | 1.196 | 1.949 | 1.822 | 1.898 | 1.226 | 2.019 | 1.878 |
| MAE [ m s$^{-1}$] | 1.395 | 0.856 | 1.474 | 1.489 | 1.400 | 0.901 | 1.533 | 1.524 |
| Bias | -0.001 | -0.007 | 0.077 | -0.110 | 0.010 | -0.003 | 0.081 | -0.112 |
| R$^2$ | 0.84 | 0.93 | 0.84 | 0.84 | 0.82 | 0.93 | 0.86 | 0.86 |
| N | 4885 | 4446 | 2576 | 2576 | 6180 | 5739 | 2576 | 2576 |

The RFM-predicted wind speed was evaluated against the wind speeds measured at FINO1 at 91 m, see Table 6. While the change in the bias is negligible, a 63% increase in RMSE is observed, which is however only 8% higher than the RMSE of the model trained and optimized at FINO1 as seen in Table 6.

The procedure was repeated by training the RFM using WT measurements at FINO3 (instead of the DMI L4 SST product) and extending it to the FINO1 site using ASCAT wind speed and direction for FINO1 while WT and all other atmospheric parameters remained the same as in the training process, i.e. as measured at FINO3. In this case, the RMSE of the extended model increases by 65% or by 4% when WT measurements from the mast site are used, compared to the DMI L4 SST. In both cases, a large increase of RMSE is seen when extending the model to the FINO1 location, although the increase in RMSE is

less when using the DMI L4 SST product.



Finally, the procedure was reversed, i.e. a model was trained using FINO1 measurements and extended to FINO3. This was performed twice, i.e. with the DMI L4 SST product and in situ measured WT. A higher RMSE was found in both cases compared to the model trained at FINO3, yet using the DMI L4 SST only increases the RMSE by 1%, with a larger associated increase in bias. Including the DMI L4 SST product to the model extension improves the prediction RMSE by 2% (Table 6)

compared to using the measured WT. In this case, a lower RMSE is obtained in extending the model to FINO3 than the other way around, even showing a lower RMSE at FINO3 when including WT measurements.

### 3.4    Spatial extension of the model

To investigate the random-forest model performance when the extension is performed over an area around the training site rather than at a single point some distance away, the RFM was trained and extended over an area using two approaches, i.e.

the WT measurements from FINO3 and the DMI L4 SST product at each WVC. Results were then compared to the NORA3 reanalysis at each WVC.

#### 3.4.1    Including in situ water temperature measurements

Initially, the RFM was extended over an area defined as 10 by 10 ASCAT wind vector cells (WVC) centered around FINO3. This was performed using WT and all atmospheric variables measured at FINO3, assuming horizontal homogeneity offshore,

while ASCAT wind speed and direction values were used at each WVC.

Figure 4a shows the 2018 mean annual wind field for the study area from the ASCAT 10 m winds, the RFM at 101 m (b) and NORA3 100 m wind speeds (c). This year was selected due to the high availability of ASCAT (MetOp-A,B,C) and NEWA data availability, as well as the low RMSE between the RFM and measurements at FINO3 (see Table 5).

A general increase in wind speed across the entire area can be observed from 10 m to 100 m, while the structure and features

of spatial variability in the wind field are not maintained. The range of RFM-predicted wind speeds across the study area varies by 0.5  m s$^{-1}$, from 8.8 to 9.4  m s$^{-1}$, while in the 10 m ASCAT wind field the speed ranges from 7.5 to 8.2  m s$^{-1}$, i.e 0.7 m s$^{-1}$. In the northeast part of the selected area, where the Horns Rev 2 & 3 wind farms are located, a smaller increase in wind speed from 10 m to 100 m is observed compared to the surrounding areas. NORA3 shows higher variability of around 1 m s$^{-1}$, from 9.0 to 10.0 m s$^{-1}$ with lower winds speeds in the south-east area and higher winds in the north-west.

The wind speed difference between the RFM and NORA3 100 m mean winds is shown in Figure 4d. Wind speed differences of -0.5  m s$^{-1}$ or larger indicate that the RFM under-predicts the mean wind field compared to NORA3, especially north of the FINO3 location. The smallest wind speed difference occurs in the south-east part of the study area, coincidentally near the HelWin wind farm. This agreement can be attributed to the lower wind speeds from NORA3 in this area and the relatively constant wind speed predicted over the entire region.



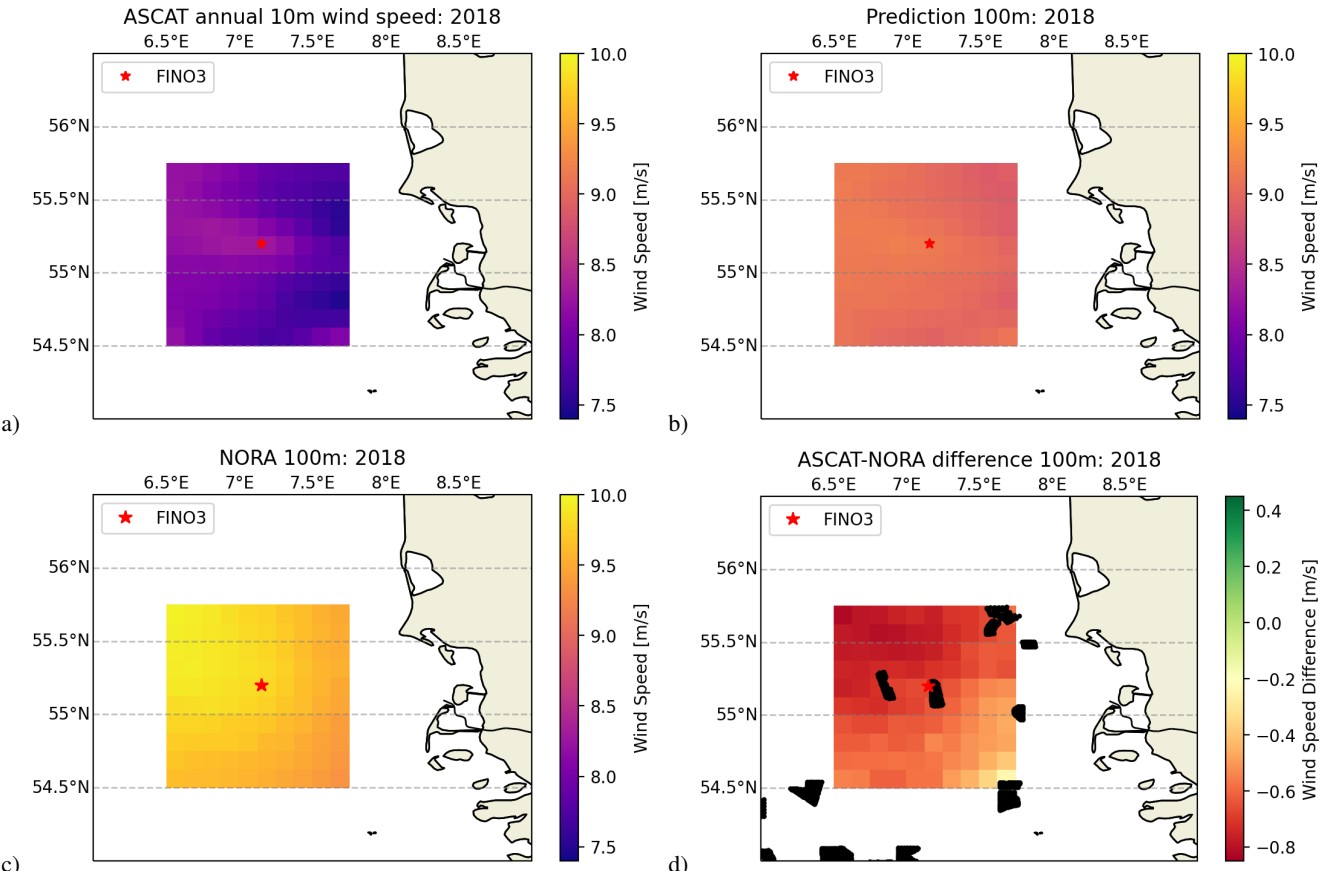

**Figure 4. a)** Mean ASCAT wind speed at 10 m for 2018 around FINO3. **b)** RFM-predicted mean wind speed at 101 m for 2018. **c)** Mean NORA3 wind speeds at 100 m regridded to the ASCAT WVCs. **d)** Wind speed difference between b) and c). The wind farm locations in the local surroundings are included in black.

### 3.4.2 Including the DMI L4 SST product

To assess the impact of SST in the spatial extension of the RFM, unique values from the DMI L4 SST product were used for each WVC along with the unique ASCAT wind speed and direction values while all other variables remained the same throughout the area of study, i.e. the measurements from FINO3. Figure 5a shows the mean SST for 2018, the mean RFM 100 m wind field using varying SST is shown in Figure 5b while the difference between RFM and NORA3 is shown in Figure 5c. The RFM wind speeds are higher than what was found when water temperature measurements from the FINO3 site were used throughout the study area, see Figure 4b, however spatial variability ranges around ∼0.3 m s$^{-1}$ across the entire region.

The difference between the RFM, using the DMI L4 SST product at each WVC, and NORA3 100 m winds (see Figure 5c) indicates a significant change compared to what was found when the measured WT was used for the RFM (see Figure 4d). The large negative differences on the north-west part of the domain are near-zero when the DMI L4 SST product is used in the





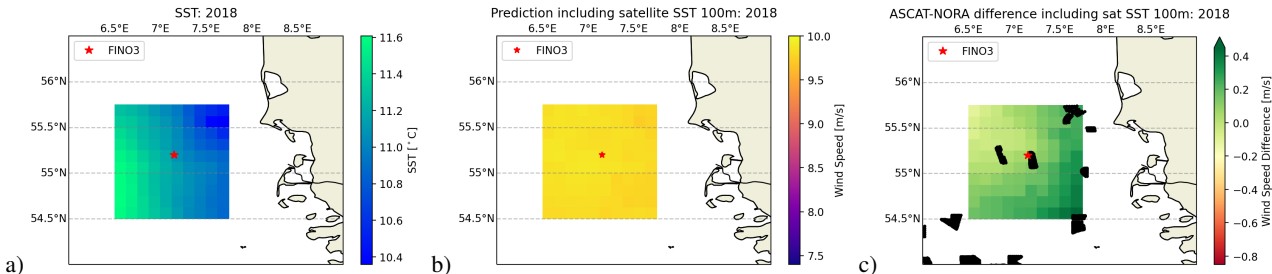

**Figure 5. a)** Mean SST for 2018 re-gridded to the ASCAT WVCs. **b)** RFM predicted wind field at 101 m with varying SST, **c)** Wind speed difference between b) and the NORA3 mean wind field at 100 m shown in Figure 4c. The wind farm locations in the local surroundings are included in black.

RFM, while areas that showed small negative biases in Figure 4d, e.g. south-east, show small positive differences of 0.3 m s$^{-1}$ indicating an over-prediction of the RFM wind speeds compared to NORA3. Contrary to what was shown in Figure 4d, the highest predicted wind speeds, and consequently lowest differences with NORA3, occur in the north-west part of the study area. The nearby wind farms are included in the plot, however there are no clear indications that they have any influence on the predictions, suggesting their contributions are negligible in the ASCAT wind retrievals.

**3.5   Data sampling characteristics**

The present study is based on training the RFM using discrete, instantaneous retrievals of wind speed and direction from ASCAT rather than the typical 10 min measured time-series used in other studies (Vassallo et al., 2020; Bodini and Optis, 2020; Optis et al., 2021). In this section, the effect of discrete sampling on the RFM training is explored utilising the 12-year long ASCAT observation period.

Figure 6 shows the number of collocated samples with the FINO3 met mast with each launch of the MetOp satellites. Since the launch of MetOp-B in 2012, a large increase into the number of samples is seen spanning the majority of the training time period.

  The number of available ASCAT observations at each WVC of the study area for the years 2010, 2018 and 2020 is shown in Figure 7. A non-uniform pattern in data availability is observed, associated with the ascending and descending orbits of

the MetOp platforms. Note that MetOp-B was launched in 2012, MetOp-C in November of 2018, while MetOp-A was de-orbited in November 2021. Hence, Figure 7a only shows observations from one instrument, while in 2018 and 2020 (b, c) two instruments were available, hence the higher range of data availability.

  To examine the impact of the sample size, the RFM was trained over different temporal periods and using varying amounts of randomly sampled subsets from the entire dataset. Figure 8a shows the RMSE (top) and bias (middle) between the RFM

and FINO3 wind speed measurements at 101 m, along with the number of samples (bottom) when training the RFM each year between 2010 and 2021 at the FINO3 site (black lines). Years were then ranked from lowest to highest RMSE for the 101 m predicted wind speeds - found in 2012 and 2019, respectively. The evaluation metrics (RMSE and bias) were calculated for the

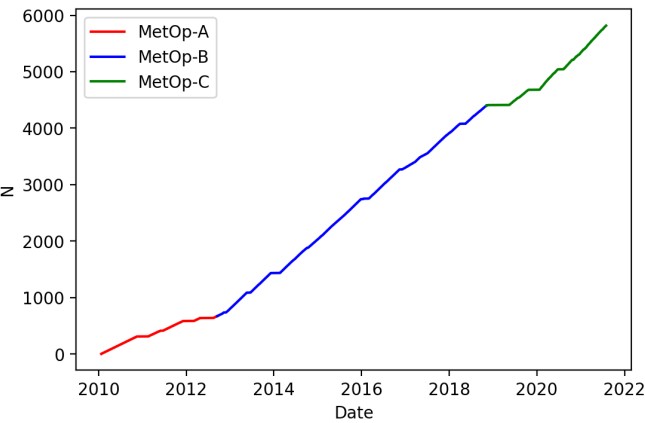

**Figure 6.** Number of concurrent ASCAT observations with the launch of each MetOp satellites at the FINO3 location from 2010-2022.

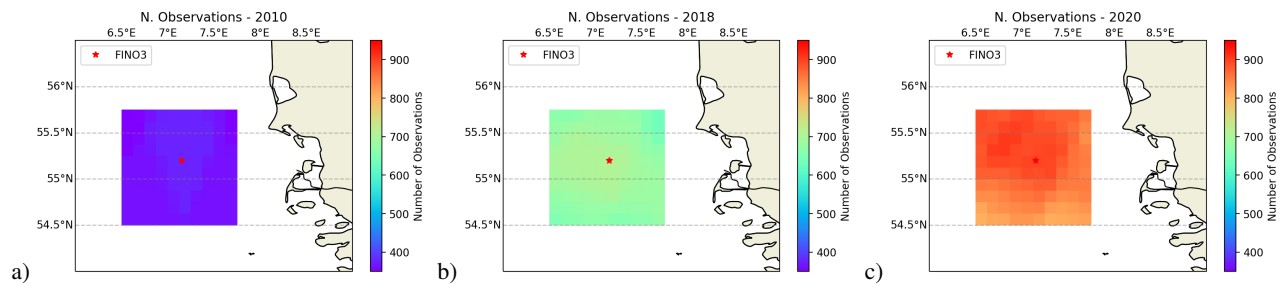

**Figure 7.** Number of available ASCAT observations at each WVC of the study area for (a) 2010, (b) 2018 and (c) 2020.

RFM trained on the best preforming year, the two best performing years, the three best, etc. The statistics, shown in Figure 8a, begin to plateau towards a stable value of RMSE and a negligible bias after 4 years of training when the sample size is around

280 2500.

The same procedure was then repeated for the RFM trained sequentially, i.e. only for 2010, for 2010-2011, 2010-2012 and up to the whole period 2010-2021. The RMSE, bias and sample size shown in Figure 8a (gray lines), indicates that although the bias converges around 4 years (or 2000 samples), the RMSE takes longer time to converge at around 6 years. Convergence of the RMSE and bias towards stable values occurs after 4 (black lines) or 6 (grey lines) years and for just over 2500 samples.

In both instances in Figure 8a does the RMSE converge around the 4-year mark between 2000-2500 samples.

Finally, the RFM was trained using random sub-samples of the full 12-year dataset instead of yearly sub-sets. The RMSE and bias between the RFM trained using random sub-samples increasing in size and FINO3 wind speed observations at 101 m are shown in Figure 8b. All metrics appear to be converging to a single value after a given amount of samples between 2500 and 3000 - although some metrics plateau around 2000 samples. Results presented here represent ten averaged instances of

training the RFM with increasing random samples.



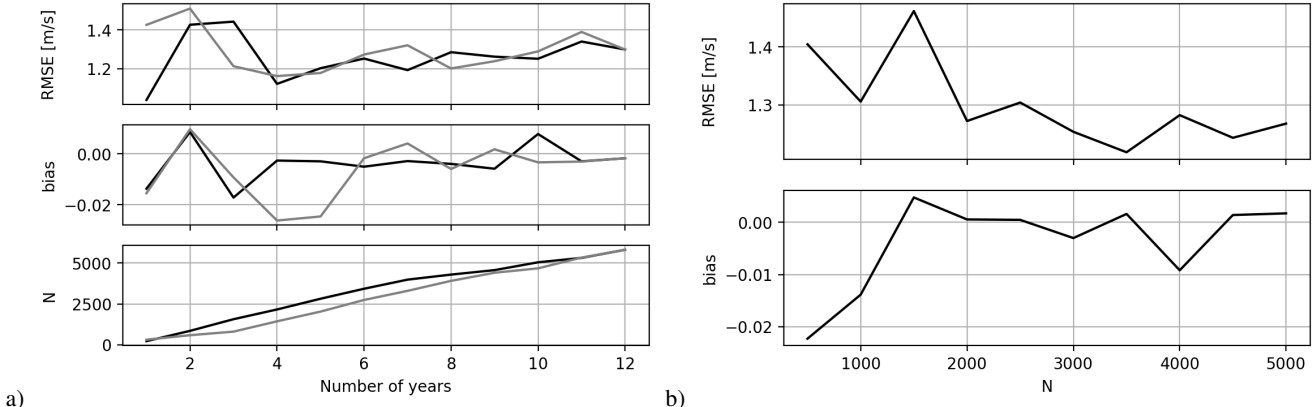

**Figure 8.** Metric evolution by an incremental number of samples used to train the RFM. a) Lowest RMSE individual years trained sequentially (black lines) and sequential training by year from 2010-2021 (gray lines), b) Training averaged cumulative random samples of the total dataset.

To investigate the impact of the sample size on the extrapolated wind speed and resulting wind profile, sub-sets of the 2018 dataset were used to estimate wind profiles shown in Figure 9a. Just as in Section 3.4, the RFM (red dashed line) over-predicts wind speeds at higher heights compared to the FINO3 measurements collocated with the ASCAT observations (red dots). In both cases, the average wind speeds are higher than those estimated from the entire FINO3 measurement period (2010-2021,

black crosses). This provides a possible explanation for the over-prediction of the RFM compared to NORA3 when trained on the subset of FINO3 data.

Figure 9b shows mean wind profiles from the RFM trained on 500 samples - around the same size as that of ASCAT in 2018 shown in Figure 9a, 2500 samples (c) and 5000 samples (d). The RFM (red dashed line) predicts higher winds speeds at heights above 70 m compared to FINO3 measurements for the case of 500 samples (red dots), while both are higher than the

complete FINO3 dataset (black crosses). Nonetheless, when increasing the sample size to 2500 (c) and 5000 (d), agreement with the corresponding FINO3 measurements significantly improved. Finally, increasing the sample size from the converging value of n=2500 to higher values, e.g. 5000, showed little to no change in the overall wind speed predictions.

Due to the sun-synchronous nature of the MetOp satellites, the FINO3 location is observed twice per day, in the morning and evening. The number of ASCAT observations as a function of the time of day is shown in Figure 10 where dark grey bars

represent the entire period and grey bars only year 2018. The majority of ASCAT observations occur between 8:00-10:00 and 19:00-21:00 with slight variations from 2018. Hourly averaged wind speed measurements from FINO3 at 107 m for the entire period 2010-2021 are shown as a dark grey line while the light gray line represents only the 2018 hourly means. At the ASCAT overpass times, i.e. 8:00-10:00 and 19:00-21:00, the collocated FINO3 mean wind speed tends to be higher than during the rest of the day, more pronounced for 2018 yet also valid for the entire 2010-2021 period. This may provide an explanation for

the RFM wind speed over-predictions compared to FINO3 and NORA3. As the RFM is trained using these higher collocated wind speeds the over-predictions may be related to the temporal sampling of ASCAT.

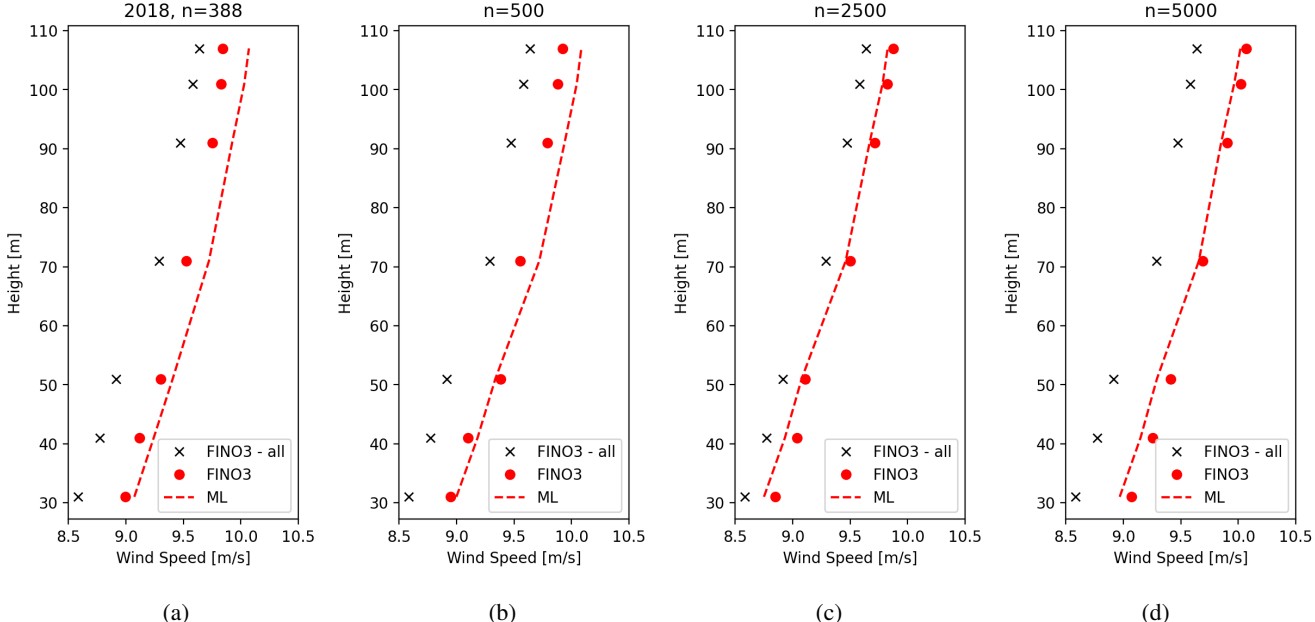

(a)          (b)          (c)          (d)

**Figure 9. a)** RFM mean wind profile for 2018 using the full dataset. **b, c, d)** RFM wind profile trained on a subset of 500, 2500 and 5000 random samples from the total dataset. Red dots represent FINO3 average wind measurements of the training dataset, black x's represent the mean FINO3 values for 2010-2021.

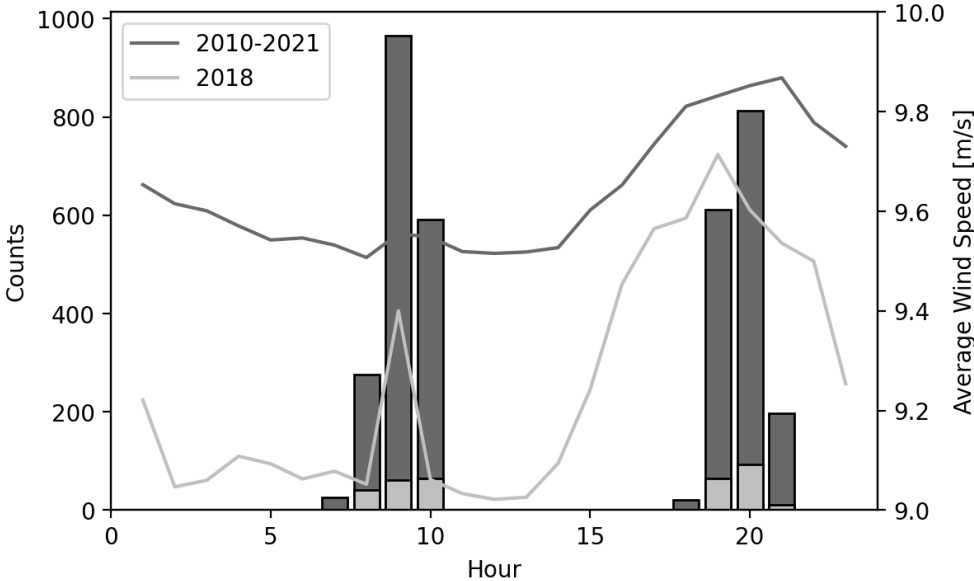

**Figure 10.** Number of hourly ASCAT observations (bars) at the FINO3 site for 2010-2021 (dark grey) and 2018 (light grey). Mean hourly FINO3 wind speed at 107 m (lines), for 2010-2021 (dark grey) and 2018 (light grey).





# 4   Discussion

This study used machine-learning methods for the extrapolation of ASCAT sea surface wind observations to higher atmospheric levels. A random forest regressor model (RFM) was trained on the near-surface ASCAT wind observations along with measurements of various atmospheric parameters to predict wind speeds at higher heights. The study area included the North and Baltic Seas, specifically the locations of the three FINO meteorological masts. For the assessment of the predicted datasets, simulated winds from NEWA and NORA3 were used. In all occasions the RFM trained at the FINO3 site out-performed the collocated NEWA WRF simulations compared to in situ measurements, with an RMSE of 1.23 m s$^{-1}$, an improvement of over 35% compared to the NEWA WRF case. Results presented in this study indicate the RFM was able to predict mean winds with a similar level of error as that of studies extrapolatinglow level winds to hub heights from met mast (Bodini and Optis, 2020), floating lidar systems (Optis et al., 2021) and with the addition of satellite data (de Montera et al., 2022).

NORA3 was selected for this study as it has been shown to represent the upper percentiles of wind speed much better than ERA5 and the older hindcast NORA10 (Haakenstad et al., 2021). Solbrekke et al. (2021) validated NORA3 against ERA5 reanalysis data where both wind speed and direction observations from six offshore sites along the Norwegian continental shelf show clear improvement over ERA5 data over both ocean and complex terrain when compared to observational wind speeds. Cheynet et al. (2022) also showed that NORA3 out-preformed the NEWA WRF dataset (Witha et al., 2019) in RMSE, bias and R$^2$ at the FINO1 met mast.

The discrepancies at heights above 51 m in RFM reconstructed wind profile may be related to atmospheric stratification, as suggested in Optis et al. (2021), where differences between predictions under unstable versus stable conditions were shown. From a similar analysis performed (not shown), results were in agreement with those in Optis et al. (2021), i.e. the RFM was able to capture the unstable profiles but over-predicted the wind profile at higher heights under stable conditions. The effects of atmospheric stability are also encapsulated in the inclusion of air-sea temperature difference as a feature for the RFM training, similar to Optis et al. (2021), which increases the RMSE by 20%. This is further emphasized when the satellite-based DMI L4 SST product was used, specifically for the round-robin comparisons and the spatial extension of the model. In both cases, comparisons with measurements from the met masts and NORA3 improved when the DMI L4 SST product was used.

The impact of including the air-sea temperature difference is evident in the training process as it completely overshadows the other atmospheric and temporal training features with the obvious exception of the satellite derived wind speed. Without including the air-sea temperature difference there is a larger contribution from the SST and air temperature while including it decreases the overall RMSE by over 20%. This was one of the main drivers for including the DMI L4 SST product to the spatial extension of the RFM. The noticeable improvement of the model compared to NORA3 is evident in Figure 5a, where the spatial variability of the mean SST field is over 1 K from the east to west, suggesting that the use of a static water temperature measurement at FINO3 is not ideal. However, wind speeds from predictions using the water temperature measurement and assuming horizontal homogeneous FINO3 atmospheric measurements were lower by on average 0.5  m s$^{-1}$ compared to NORA3, see Figure 4d, suggesting that the assumption may still be valid offshore at these distances from the coastline.



One parameter not considered in previous studies, e.g. Bodini and Optis (2020); Optis et al. (2021), was the length of the training period where it ranged from a few months to a few years of in-situ mast or lidar datasets, typically consisting of 10 min measurements. This study uses a discrete subset of satellite wind retrievals and although it covers a longer period, the number of available observations is smaller compared to 10 min datasets even if the latter extend over shorter periods. Therefore it was considered important to evaluate the model trained over different periods of time. From results presented here, training statistics

converged when the sample size increased, reaching a plateau after approximately 2500 samples, suggesting this as a minimum number of samples to properly train a RFM when using satellite observations. This is consistent with findings from Barthelmie and Pryor (2003) where 2000 satellite observations were considered sufficient to represent wind resource statistics. Given the required data availability, only scatterometer winds were used to train the model. SAR winds have higher spatial resolution, nonetheless their data availability is reduced due to a lower temporal sampling frequency ($\sim$ 3 days). Nonetheless, for areas

were SAR winds offer a significant sampling coverage, it would be relevant in a future study to examine their applicability for training RFMs and extrapolating surface winds to higher atmospheric levels.

de Montera et al. (2022) addresses the sampling problem with the lack of SAR images (500 samples in their study) with simulating satellite passes with WRF outputs. Similar to this, the RFM method could be applied with supplementary scatterometer data from other missions together with ASCAT. This is expected to provide more robust results from the RFM

method. Currently operating missions are HY-2B and HY-2C (Haiyang satellites) with the HSCAT scatterometer instrument onboard launched by the Chinese National Satellite Ocean Application Service (NSOAS) (Zhao et al. (2021)). The China-France Oceanography Satellite CFOSAT satellite with a scatterometer launched by Centre National d'Etudes Spatiales (CNES) and China National Space Administration (CNSA) is in operation. CFOSAT winds have been compared to buoy data (Zhu et al. (2022)). The Indian Mini Satellite with SCATSAT-1 scatterometer onboard launched by Indian Space Research Organisation

(ISRO) is in operation (Misra et al. (2019)). Furthermore, archived data from past missions could be considered such as HY-2A from NSOAS, the ScatSat-1 satellite with the OSCAT scatterometer onboard launched by the ISRO (Wang et al. (2019)) and the American QuikSCAT satellite with the SeaWinds scatterometer onboard launched by the National Aeronautics and Space Administration (NASA). QuikSCAT observations have been used for wind resource mapping (Karagali et al. (2014)). Additional samples from other missions would increase the number of samples and would fill-in at other times of the diurnal cycle

thanks to different orbital paths than ASCAT.

The features used in the machine-learning training process were selected based on their availability for applying the training approach to floating lidar systems in deep-sea environments, since all atmospheric measurements are readily available on current floating lidar systems or through satellite data. Offshore floating lidar systems only provide vertical wind measurements at specific locations, similar to meteorological masts, therefore spatially extending such measurements using 2-d satellite wind

fields and machine-learning methods is of great interest.

Nonetheless, the need for a large enough sample size of at least 2500 discrete observations may be a limiting factor as floating lidar systems are typically deployed for periods of 1 to 2 years or less and would not yield the proposed number of collocated observations with the current ASCAT instruments as can be seen in Figure 6. Ferry mounted lidar systems (Gottschall et al., 2018) have been compared with ASCAT winds (Hatfield et al., 2022); they can also provide spatial sampling not achieved when



measurement systems are moored at specific locations. Although the dataset used in Gottschall et al. (2018) covered a period
of only 5 months, the concept involves mounting lidar systems on established ferry routes thus providing the opportunity
for longer time-series measurements over established paths. Having lidar systems alongside the corresponding atmospheric
sensors on already established ferry routes could provide long-term measurements in deep water areas suitable for training a
machine-learning model.

With the application of satellite wind retrievals in machine-learning predictions of long-term mean wind speed estimates,
the discrete nature of the observations needs to be considered. For the time interval 18:00-21:00, when ASCAT has the highest
sample availability, shown by the bars in Figure 10, mean winds measured at FINO3 are higher compared to the rest of the day,
more pronounced for 2018, as seen in Figure 10 (lines). This suggests that the temporal dependence of sampling availability
may influence the RFM comparisons with NORA3 and in situ measurements at FINO3, especially when limited comparison
periods are considered (2018) as artefacts can be introduced because the trained dataset includes features and variability that
are not necessarily present during the specific period of comparison. This can potentially explain the over-estimation of RFM
predicted winds compared to NORA3 and FINO3 measurements at all heights in Figures 4d and 5c. This is further supported
by results shown in Figure 9 where for profiles using lower sample sizes, as in 9a, an over-prediction of both the RFM (red
dashed line) and the F3 measurements (red dots) is found compared to the profile using all available measurements at FINO3
(crosses).

Bodini and Optis (2020) outlined the importance of applying a round-robin approach when validating models trained in one
location to another. While using machine-learning models where hub-height relevant wind measurement are known may not
be of interest, extending those to the area surrounding the training site is of interest as it can provide a better description of the
ambient wind field. In this study, this approach was applied between the FINO1 and FINO3 met masts (as outlined in Table
6). In all cases, a model trained at FINO3 out-performed that at FINO1 in all evaluation metrics. The same result is seen in the
comparisons with the NEWA data, see Table 5 and Witha et al. (2019), as well as with the NORA3 data, having an RMSE of
0.8 m s$^{-1}$ at FINO3 and 1.3 m s$^{-1}$ with FINO1 (Cheynet et al., 2022). This could be attributed to the proximity of FINO1 to
land (45 km) or the high density of wind farms. With a westerly-dominated wind direction and located directly in the BorWin
wind farms, the wind farm wakes could affect the wind speed measurements at 91 m, having no free stream wind profiles.

Extending the model spatially and evaluating the results with NORA3 in Figures 4 and 5 shows that including the satellite
SST greatly improve the results. However, in both figures, the RFM was not able to fully reproduce the spatial wind structure
as shown in the NORA3 data (Figure 4c). Both figures show a resemblance to the ASCAT 10 m wind speeds (Figure 4) but
with a much narrower range of wind speeds (0.5 m s$^{-1}$ and 0.3 m s$^{-1}$ respectively), where the 10 m wind speed distribution
should not be entirely representative of that at 100 m especially in different atmospheric stability regimes. It can also be noticed
that in the ASCAT wind retrievals, in the WVCs enveloping the nearby wind farms, a slightly higher wind speed is observed.
This can be attributed to higher reflection caused by the wind farms leading to higher wind retrievals. This can directly impact
the RFM as in both Figure 4d and 5c the highest wind speed difference with NORA3 is found in the bottom-right WVC, an
area with a wind farm and a higher wind speed at 10 m from ASCAT.



# 5 Conclusions

The aim of this study was to explore the applicability of machine learning methods for training a model to extrapolate ocean surface wind measurements from satellites to higher atmospheric levels.

Using a random forest model approach it was possible to effectively recreate the vertical wind profile at FINO3 with only slight over-predictions at the higher atmospheric levels, i.e. between 0.03-0.07 m s$^{-1}$. A similar pattern was observed when the model was extended over an area of 125 m$^2$ surrounding the FINO3 mast. The RFM was found to over-predict the wind speed when compared to the NORA3 re-analysis data over the same area, however including satellite-based SST retrievals over the entire area into the training dataset improved the agreement.

Special attention should be given to the training procedure when using observations with a limited daily temporal resolution, e.g. 2-4 times per day, as training datasets, such as ASCAT. In those cases, over/under-prediction of the parameter of interest compared to simulations or in situ measurements may result from the sampling of the original training dataset, regardless of the number of samples used in the training process.

Results from this study show the prospect of applying machine-learning methods for the purpose of extrapolating surface winds to higher atmospheric levels. An interesting application of such methods is to datasets from offshore floating lidar systems (floating lidar systems) specifically for their extension from point measurements to other locations within the area of interest. Such applications would require the availability of measurements spanning at least 2-3 years with the concurrent ASCAT daily coverage. Extending the period of coverage will not only benefit the available collocated measurements and thus the machine-learning statistics, but will also provide a more representative time period for wind resource assessment than the typical 1 to 2 year time scales.

*Data availability.* The NORA3 is published at https://thredds.met.no/thredds/catalog/nora3/catalog.html (last access: August; NORA3, 2022). The DMI SST dataset can be obtained from http://marine.copernicus.eu/ (last access: 25 February 2022; Copernicus marine service, 2022). The ASCAT data was taken from https://marine.copernicus.eu (last access: August 2022) The FINO data can be obtained from http://fino.bsh.de (last access: August; FINO, 2022).

*Author contributions.* D.H. prepared the original draft, as well as acquired, developed, and performed the data analysis and produced the results. C.B.H. and I.K. contributed in numerous discussions, provided suggestions, and supported the interpretation of the results. I.K: provided the satellite SST data, wrote the SST section and provided text and edits for the results and discussion sections. C.B.H provided text for parts of the discussion. All authors reviewed and edited the manuscript until it reached the final stage. All authors have read and agreed to the published version of the manuscript.

*Competing interests.* The authors declare no conflicts of interest.



*Acknowledgements.* This project has received funding from the European Union's Horizon 2020 research and innovation programme under the Marie Sklodowska-Curie grant agreement number 860879.



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
