# Peer review of "Vertical extrapolation of ASCAT ocean surface winds using machine learning techniques"

_Wind Energy Science, 2022_

## Referee Comment (RC1)

**REVIEW: Vertical extrapolation of ASCAT ocean surface winds using machine learning techniques**
*Hatfield et al.*

**General comment:**

The paper describes the application of machine learning algorithms to vertically extrapolate near-surface winds derived from satellites data. The focus is offshore, in Northern Europe.

The paper is generally well written (although some grammar improvements here and there would be recommended), and it represents a nice academic application of machine learning to the wind energy sector. However, I question the practical utility of this approach in real world applications. The main reason for this is that satellite-derived observations of wind speed at a given location are only available ~ twice a day, and always at around the same times. I am really struggling in finding a situation where someone would be interested in knowing hub-height wind speed only at these two hours of the day. While practical applicability is not a strict requirement for having a paper published, I still think this limit should be at the very minimum highly stressed in the paper, and language softened to reflect the limited applicability of the results described in the analysis.

**Specific comments:**
1. L.19: add "above the sea surface" or something similar.
2. L. 26: 275 m seems like a very specific threshold – can you provide a reference?
3. L.45: "predicting" instead of "predict" (or "to predict")
4. L. 66: "higher" instead of "greater"
5. L. 106: in the text you mention water temperature, while in Table 1 you mention sea surface temperature; please be consistent.
6. L.111: do you mean that all variables have exactly an availability of 85%, or greater than 85%?
7. L.123: what's the temporal resolution?
8. Consider moving all the details regarding data access to the 'Data availability' section towards the end.
9. Figure 1 caption: "shapes" instead of "rectangles".
10. It is hard to fully understand Table 3 (and some of the discussion in this section) without a clear explanation of the temporal frequencies considered here for the various data sources. Does "total data" refer to 30-min average time periods? And "Concurrent data with ASCAT" to 30-min average time periods in which an ASCAT data point was recorded? Please clarify in the paper.
11. Table 2: "FINO" is repeated twice in the left column.
12. Why using the cosine of wind direction only (and not the sine, too)?
13. Once again, being clear about the temporal resolution of the data used is key to understand whether the random split between train and test sets is a right choice, or autocorrelation effects might play a role in artificially enhancing the ML results.

14. Table 5: I would argue that the most relevant comparison is ML vs NEWA at each site, so I would suggest adding a horizontal line after each site, and highlight in bold the "winner" metrics at each site.
15. L.170: clarify that the error values refer to the test set.
16. L.173: "Note" instead of "not".
17. Section 3.2: please clarify how the mean wind profile from the RF was computed. Do you simply apply the RF to the whole period, and average results? Or to the test set only?
18. Section 3.3: why not including FINO2 as well?
19. Section 3.4: can you somewhat verify your hypothesis of horizontal homogeneity by looking at spatial variability of the meteorological variables from NEWA and the reanalysis product?
20. Figure 4: once again, more clarity is needed when explaining what is being plotted and described. What's the temporal extent of what is shown? Are NORA data taken only at time stamps at which ASCAT data are available? How about the ML-extrapolated winds? Are we really comparing apples to apples? Once that is clarified, please adjust text accordingly (it is misleading to state you are showing 2018-averaged winds, if you are only cherry picking time stamps).
21. Figure 4: why wasn't NEWA included?
22. Conversely, why wasn't NORA3 included in the earlier analysis (Table 5)? It is essential to know how well it compares to observed winds in order to use it as proxy for the truth here.
23. Figure 6: please change labels and instead list ALL the satellites available in each time period. Also, specify that this is a cumulative plot. What do you mean by "concurrent" in the caption?
24. Discussion of Figure 10 is key (see my main major comment above), and in my opinion should be moved way earlier in the paper.
25. Figure 10 caption: specify when referring to bars vs lines. Also, specify you are referring to local time and not UTC time (I believe).
26. L.332: decreases instead of increases?
27. Data availability: why not sharing the model algorithm scripts as well?
28. Please double check references and make sure each has a DOI.

---

## Author Comment (AC1)

**Response to Referee #1**

**Daniel Hatfield, Charlotte Bay Hasager, Ioanna Karagali**

**February 14, 2023**

First of all, thank you very much for taking the time to review this article and for your positive comments and suggestions. Below are responses (in black) to all of the referee's comments (in blue).

**General Comments**

"The paper describes the application of machine learning algorithms to vertically extrapolate near surface winds derived from satellites data. The focus is offshore, in Northern Europe. The paper is generally well written (although some grammar improvements here and there would be recommended), and it represents a nice academic application of machine learning to the wind energy sector. However, I question the practical utility of this approach in real world applications. The main reason for this is that satellite-derived observations of wind speed at a given location are only available twice a day, and always at around the same times. I am really struggling in finding a situation where someone would be interested in knowing hub-height wind speed only at these two hours of the day. While practical applicability is not a strict requirement for having a paper published, I still think this limit should be at the very minimum highly stressed in the paper, and language softened to reflect the limited applicability of the results described in the analysis."

Thank you very much for you comments and attention to detail while reviewing the manuscript. We agree that the practical application of the methods presented in the study are limited, but would like to remind the referee that this was a *proof of concept* style of paper, exploring the potential of using machine-learning method as a way of extrapolating satellite data as opposed to the alternatives that have not had much success. This was done in hopes to further utilize the invaluable daily satellite surface wind coverage for longer-period wind resource assessment.

We do agree that the language used in the paper needs to be "softened", in which sections and word-choices have been changed to stress the limitations/potential of applying machine learning methods.

**Specific Comments**

1. L.19: add "above the sea surface" or something similar. Added

2. L. 26: 275 m seems like a very specific threshold – can you provide a reference? We have included references for both ferry and buoy based lidars with heights up to 275m in line 26 as (Rubio et al. 2022 and Hatfield et al. 2022)

3. L.45: "predicting" instead of "predict" (or "to predict") Added

4. L. 66: "higher" instead of "greater" Added

5. L. 106: in the text you mention water temperature, while in Table 1 you mention sea surface temperature; please be consistent. This has been changed in Table 1 to Water Temperature

6. L.111: do you mean that all variables have exactly an availability of 85%, or greater than 85%? We have clarified this in line 113 "All measured quantities show a data availability above 85% with the exception of WT (76%)"

7. **L.123: what's the temporal resolution?** The temporal resolution is *daily* coverage (which is now clarified in the text in line 121). With the high heat capacity of water small diurnal changes are observed and uncertainties are provide by Høyer & Karagali 2016 (also mentioned in the text). It should be noted that the SST product has a mean difference of -0.06°C compared to moored buoys and a 0.46°C standard deviation of the differences.

8. **Consider moving all the details regarding data access to the 'Data availability' section towards the end.** Thank you for the suggestion, we will however include the data in the text to remain consistent with other papers using the satellite/FINO data.

9. **Figure 1 caption: "shapes" instead of "rectangles".** Added

10. **It is hard to fully understand Table 3 (and some of the discussion in this section) without a clear explanation of the temporal frequencies considered here for the various data sources. Does "total data" refer to 30-min average time periods? And "Concurrent data with ASCAT" to 30-min average time periods in which an ASCAT data point was recorded? Please clarify in the paper.** This is correct, "total data" refer to 30-min average time periods at the masts, "concurrent data with ASCAT" refers to 30-min average time periods in which an ASCAT data point was recorded. This is now updated in the Table 3 label.

11. **Table 2: "FINO" is repeated twice in the left column.** Removed

12. **Why using the cosine of wind direction only (and not the sine, too)?** We have used both at one point in the early stages of this work where using the cosine of the wind speed performed better in the ML training (resulting in lower RMSE).

13. **Once again, being clear about the temporal resolution of the data used is key to understand whether the random split between train and test sets is a right choice, or auto-correlation effects might play a role in artificially enhancing the ML results.** Added in Table 2 caption as: "All of the data measured from the FINO masts are 30 minute averaged."

14. **Table 5: I would argue that the most relevant comparison is ML vs NEWA at each site, so I would suggest adding a horizontal line after each site, and highlight in bold the "winner" metrics at each site.** This is a very good suggestion and is now incorporated in Table 5 along with the addition of the same comparison with NORA3 at FINO for the same periods as NEWA.

15. **L.170: clarify that the error values refer to the test set.** This is clarified in line 171 as "...with the test dataset..."

16. **L.173: "Note" instead of "not".** Added

17. **Section 3.2: please clarify how the mean wind profile from the RF was computed. Do you simply apply the RF to the whole period, and average results? Or to the test set only?** This is now clarified in the text in line 172: "The model trained in Section 3.1 is applied to the entire 12-year collocated dataset at all heights at FINO3 from 31 m to 107 m."

18. **Section 3.3: why not including FINO2 as well?** FINO2 is much too far away and the Baltic sea has a very different marine atmosphere. As this is still an article exploring the *proof-of-concept*, we have limited the spatial extension / round-robin approach to the North Sea. It should also be noted in the ML papers extrapolating wind speeds (i.e. Optis et al. (2021) and Bodini et al. (2020)) that use the round-robin approach, their distance are less than 100km apart, whereas the distance between FINO1 and FINO3 already exceeds that of previous work (136km). Thus, the comparison with FINO2 is beyond the scope of this article.

19. **Section 3.4: can you somewhat verify your hypothesis of horizontal homogeneity by looking at spatial variability of the meteorological variables from NEWA and the reanalysis product?** In Figures

4a and 5a we can see the spatial variation of both the wind speed (from NORA3) and SST (from DMI L4 SST) both with very low variation across the 125km$^2$ area. The FINO3 area is also in open ocean, far from the coast and without islands within the study area. We think this is a very fair assumption to make. We have also verified using ERA5 data for the North Sea that there is small variations in air temperature ($\pm 1°$C similar to that of the SST variation) across the chosen area which is not shown.

20. Figure 4: once again, more clarity is needed when explaining what is being plotted and described. What's the temporal extent of what is shown? Are NORA data taken only at time stamps at which ASCAT data are available? How about the ML-extrapolated winds? Are we really comparing apples to apples? Once that is clarified, please adjust text accordingly (it is misleading to state you are showing 2018-averaged winds, if you are only cherry picking time stamps). ASCAT is the limiting feature, all time-stamps need to be concurrent with ASCAT. This is however is not the case in Figures 4 & 5, these included yearly averaged 30 minute NORA3 grids and have now been updated to be concurrent with ASCAT so that we are comparing "apples to apples". The Figures 4 & 5 are now updated to include concurrent NORA3 data instead of yearly averaged for 2018.
Thank you very much for the comment!

21. Figure 4: why wasn't NEWA included? Cheynet et al. (2022) has shown that NORA3 out-performs the NEWA at FINO1 but this is not mentioned until the Discussion. A clarification is now added in lines 228 "It should be noted that only NORA3 will be included in the spatial comparison with the RFM as it has out-performed NEWA at the FINO3 mast in Table 5 and in Cheynet et al. (2022) at FINO1.". NORA3 comparison was also added in Table. 5 to further this argument.

22. Conversely, why wasn't NORA3 included in the earlier analysis (Table 5)? It is essential to know how well it compares to observed winds in order to use it as proxy for the truth here. Added

23. Figure 6: please change labels and instead list ALL the satellites available in each time period. Also, specify that this is a cumulative plot. What do you mean by "concurrent" in the caption? Concurrent is confusing, it is now changed to "Cumulative number of samples of ASCAT observations at the FINO3 location from 2010-2022. The vertical lines represent the launch of each MetOp satellite as well as the decomission date of MetOp-A." in the Figure 6 label. The Figure is now updated to show which satellites were also in operation during this time period.

24. Discussion of Figure 10 is key (see my main major comment above), and in my opinion should be moved way earlier in the paper. I agree that this is a major point of discussion, but we do think it is in the right place within the results. This paper explores the implications of the use of ML on satellite extrapolation where we explore vertically (wind profile), horizontally (round-robin) such as previous work but also the spatial (in comparison with NORA3) and temporal (in sampling) domains due to the nature of satellites. We completely agree that the discrete nature of the satellites is one, if not, the largest limiting factor within this paper, however we think that is explored in the discussion. Whereas, the layout of the paper slowly builds on expanding the simple vertical extrapolation of the satellite at each FINO mast.

25. Figure 10 caption: specify when referring to bars vs lines. Also, specify you are referring to local time and not UTC time (I believe). All times throughout the paper have been recorded in UTC (ASCAT, FINO, SST, NORA3) which we have added in the Data section in line 146 as "all data used from all sources is recorded in Coordinated Universal Time (UTC)". But we have added in the Figure for clarity and clarified the bars/lines.

26. L.332: decreases instead of increases? Yes, changed.

27. Data availability: why not sharing the model algorithm scripts as well? Unfortunately, there is no intention from the authors to publish the scripts as well.

28. Please double check references and make sure each has a DOI. Done

---

## Author Comment (AC2)

**Response to Referee #2**

Daniel Hatfield, Charlotte Bay Hasager, Ioanna Karagali

February 14, 2023

First of all, thank you very much for taking the time to review this article and for your positive comments and suggestions. Below are responses (in black) to all of the referee's comments (in blue).

The authors test a new model, based on machine learning (ML), to extrapolate ocean surface winds to hub heights of offshore wind masts, which is of relevance for the operation of offshore wind farms. Scatterometer ocean surface winds together with air-sea temperature differences appear the most important parameters for training the ML model. The model was trained for different time periods and the verification with independent data (not used for training) shows that the ML model outperforms an NWP model based on WRF.

**General Comments**

Although the authors have demonstrated that ML techniques can be used to extrapolate ocean surface winds to 100 meter altitude, they have not demonstrated that the methodology outperforms the use of already available NWP models.

We would like to remind the referee of the purpose of the paper in line 66: "The aim of this study is to assess the potential of using machine learning models with two-dimensional wind field observations at lower atmospheric levels in order to predict the wind at higher heights". The goal was obviously to try to out-perform model data but this is a *proof-of-concept* paper exploring the potential of satellite extrapolation through ML techniques vertically, horizontally, spatially and temporally. This method may not have out-performed the state-of-the-art NORA3 model, but we do believe that this article is still relevant for the area of wind energy and satellite wind retrievals.

It would be very helpful to shortly outline current operational practice of operators of wind farms. ASCAT winds and SST are freely available, so a ML model using these two input parameters could be an interesting option for operators in case they have no access to actual mesoscale model data. But is that the case? Although WRF is freely available as well, the NEWA dataset is of limited use for operators as it needs ERA5 as hosting model. However, ERA5 availability is at best a couple of days behind real time, and as such the NEWA approach of limited value for daily operations.

This method (as well as satellite wind observations) are more applicable in terms of wind resource assessment as opposed to forecasting or daily operational use in wind farms. Yes, we agree that the NEWA (WRF) approach is limited for daily operations, that is why in every comparison with NEWA (and for that matter NORA3) we have compared long-term statistics.

In the context of the above, can the authors please explain the relevance of the use of NEWA in their study? Also given that NORA3 outperforms NEWA WRF (line 326)

NEWA is well studied and provides a larger domain than that of NORA3. NORA3 does outperform NEWA and falls within the domain of interest in this work and therefore the inclusion of NEWA was to strengthen the argument for the comparison with NORA3.

The comparison against NEWA (WRF based) is not fair in the sense that NEWA does not make use of scatterometer winds explicitly (NEWA has no data assimilation), but only implicitly through the boundaries of the hosting model. A more fair comparison would be to use NORA3 in Table 5, because it outperforms NEWA as stated by the authors and probably makes explicit use of scatterometer winds in the reanalysis, although this was not mentioned by the authors.

NEWA was used in previous study in conjunction with ASCAT (in Hatfield et al. 2022) and was initially used in the ML comparisons until NORA3 was realized to have a better comparison. NEWA has been well study and used in many papers cited throughout this work, whereas NORA3 has few in comparison. We felt as though we needed to justify the use of NORA3 as a comparison, which is the initial idea behind the use of NEWA. The first comparison with the ML models and the FINO masts in Table 5 with the addition of a NORA3 comparison could help in further justify the use of NORA3 and the reason why NEWA is no longer used throughout the study.

"[NORA3] ... makes explicit use of scatterometer winds in the reanalysis, although this was not mentioned by the authors"

NORA3 "runs explicitly resolved deep convection and yields hindcast fields that realistically downscale the ERA5 reanalysis." (Haakenstad et al. 2021) whereas NEWA uses ERA5 for dynamical forcing (Dorenkamper et al. 2020) so both use ERA5 in the model chain. But even so, the use of satellite wind observations is more to do with the wind resource than forecasting. The two simulated datasets may also have different use cases, however they are still a large wind dataset that is used over a 12-year period which provides similar domains and resolutions.

**Major Comments**

1. Although it was not the ultimate goal of the study to test if a model based on ML does outperform an NWP model, it is of importance to operators of offshore wind farms to know if ML outperforms a mesoscale NWP model. NORA3 represented the latter, so please add NORA3 to Table 5.

The ultimate goal of the paper was to (line 66) "The aim of this study is to assess the potential of using machine learning models with two-dimensional wind field observations at lower atmospheric levels in order to predict the wind at higher heights". The use of NWP was a means of evaluation/validation due to the lack of offshore wind observations other than model data.

As to the addition of NORA3 to Table 5, we completely agree, it is a little pointless to have just the NEWA comparison. It also justifies the use of only NORA3 in the spatial extension of the ML model. This is now added.

2. Table 5. The column denoted 'N' shows for ML the number for "Concurrent data with ASCAT" (although the numbers are not exactly the same with those in Table 3). This is misleading as the numbers in the other column (RMSE, bias, ..) are based on "Data used for validation". Please use the correct numbers in Table 5.

Thank you for pointing this out. At the FINO1 site there are 6180 data points concurrent with ASCAT (as stated in Table. 5 & 6) this was double-checked and changed in Table 3. The $N = 5739$ for FINO3 is consistent across all three tables. As to the N-values in Table 5. We will leave them as is. The point of Table 3 was to demonstrate that 4/5 of N was used to train the model and the other 1/5 was used as evaluation.

3. Section 2.5. As a non-expert in ML techniques, this section was too abstract and hard to read and understand. It would help to relate the parameters in Table 2 to, X,Y and T in the text. A formula would help (see below). How does y_overbar relate to the parameters in Table 2? Is it wind speed at e.g. 100m?

This is a good point. We have related the RFM variable with the inputs and outputs used in the paper in line 154: "In the case of this study, $\overline{y}$ will be the predicted wind speed at higher heights (107 m at FINO3 for example), where $Y$ will be the concurrent wind speed measurements at the mast at the desired height." and in line 154 "... input data ($X$ in the equation above)". The only variable that does not related to data in this study is $T$, the trees, which is inherent to the RFM itself.

4. The paragraph on hyper-parameters and K-fold cannot be understood without any background knowledge of these techniques. For me it was totally unclear. We would remove it from the text. The last paragraph in section 3.1 concludes that wind at height is mainly modelled through wind at the surface (WS) and the air-sea temperature difference (AT-SST). In formula: ML(j,FINOi) = a(i,j)WS + b(i,j)(AT-SST), with j denoting altitude and i the FINOi (I=1,2,3) station. The training set then aims to estimate a(i,j) and b(i,j). Is that right?

We agree, it is worth mentioning the hyperparameters and the ranges evaluate for the RFM optimization, but the explanation of the K-fold method may be too much. We have removed section on the k-fold method leaving the necessary information of the Hyperparameters:

"While model parameters are "learned" during the training phase, *hyper-parameters* are set before the training to create a more accurate algorithm. Hyper-parameter tuning relies on experimental results of combinations of model parameters to evaluate the performance of each model.  The hyper-parameters are varied and their associated ranges are outlined in Table 4."

5. (See also remark to section 2.5 above). Line 426: "Results from this study show the prospect of applying machine-learning methods for the purpose of extrapolating surface winds to higher atmospheric levels".I think this statement is too strong as the study does not show that RFM outperforms mesoscale models which assimilate ASCAT. Please correct.

We agree the language throughout the text needs to be softened and has been. In this particular example we have change the word "prospect" to "potential" as this work lays the groundwork for the potential of extrapolation through the use of machine-learning algorithms.

6. I can imagine a seasonal dependence of the ML model parameters for the different station locations. Was this tested? Please comment.

Interesting point. We have looked at monthly averaged spatial extensions but decided that the number of samples was too small and that yearly spatial comparisons were more realistic and inline with other satellite extrapolation methods (i.e. see Badger et al. 2015 and Karagali et al. 2018 using the long-term stability correction). Optis et al. 2021 show the dependency of stability conditions on the ML extrapolation of low level lidar winds measurements (lower RMSE in unstable conditions compared to stable conditions) which is a seasonal phenomena. Thus we would expect, for example, the model to perform better in the Autumn months (predominantly unstable conditions) than in the Spring months (predominantly stable conditions).

**Minor Comments**

Line 91. "This 12.5 km product has a standard deviation of 1.7 m s-1 and a bias of 0.02 m s-1 (Verhoef and Stoffelen, 2019)." I guess this is for wind speed, not the wind components? Please make clear in the text.

It is now clarified

Line 144; why 3x3 grid. Given the NORA3 2 km grid size and ASCAT 12.5 km product, I would expect 6x6, since 6*2=12, which is close to the 12.5 km ASCAT footprint. Please explain the choice for the number 3.

NORA3 is a 3 km grid size hence the choice of a 3x3 grid. This is outlined in section 2.4 in line 134.

In Table 3, how were the "Data used for validation" selected? Randomly from the "Concurrent data with ASCAT"?

This is a good question and is now clarified in the text. The "Data used for validation" is 1/5 of the "Concurrent data with ASCAT" whereas the "Data used for model training" is 4/5. These are randomly chosen 4/5 and 1/5 splits as we tried to explain through the k-fold method, but our explanation of the k-fold method was unclear in your Major Comments, hopefully it is now clearer.

Figure 2, right panel. Why does the number 1148 differ from 1147 in Table 3? Please correct.

Corrected

Figure 5.These numbers are based on validation data only, so N=1148 (or 1147), right?

For both Figure 4b and Figure 5b, the model that was trained/validated (4591/1148) was then applied to the data for 2018; this includes the ASCAT 10m wind speeds (over N=750, Figure 7b), the FINO low-level atmospheric data and the FINO/satellite SST values. Basically, the model that was trained earlier was then applied to the 2018 data alone. We have now tried to explain this in the text that

introduces these figures.

Yes, good spot. We have re-run these to verify and it is now corrected.

The first (i.e. 63%) shows the spatial increase in the RMSE of the model trained at FINO3 whereas the second one shows the increase in RMSE from a model that was trained at FINO1. Although we see a large increase in the RMSE from extending the model from FINO3 (63%) we are only slightly increasing the error from a model optimized for FINO1 (8%) suggesting either the FINO1 91m measurements could be flow distorted (which can also be seen in the higher RMSE with NEWA or in validation such as in Witha et al 2018) or that the ASCAT gird cell could be influence from the high density of wind turbines (Figure 1) giving higher scatter - both of which are interesting consequences. We do agree with the referee that the comparison with the model trained at FINO1, for example, is more relevant. The comparison with the FINO3 model is now removed and the percentage changes mentioned above are now included in Table 6 for clarity.

Table 6 now includes the percentage changes and these 1% and 2% (FINO1 traning model extended to the FINO3 location compared to the original model trained at FINO1) are irrelevant and removed from the text.

No this was not expected. Similar wind fields to the 10m observations are seen but at a much smaller scale as seen in Figure 4b & 5b (changes in wind speeds of around 0.3 m/s across the entire 125km$^2$ area). Even with the spatial SST addition to the training (i.e. Figure 5a) there is very small changes across the entire 125km$^2$ area. This suggests the constants or unchanging data used the in the model training (i.e. air temperature, pressure, humidity) from the FINO3 mast may have a larger influence on the extrapolation than expected.

Corrected

This is true, and has been changed in line 345 as "An improvement of the RFM model taking NORA3 as reference is seen in Figure ...". It should be noted that we have tried to soften the language throughout the text to not over-exaggerate the results of the RFM.

Corrected, thank you

**Typos**

All typos were fixed that were noted by the referee, thank you for taking the time to read the manuscript so carefully.

---

## Author Response (AR2)

**Response to the associate editor**

Daniel Hatfield, Charlotte Bay Hasager, Ioanna Karagali

March 6, 2023

The manuscript proposes the use of ASCAT satellite winds, combined with observations from a few existing offshore tall towers, to train a ML method to give the vertical profile of wind speed. The approach is innovative and the results are promising, thus the paper deserves publication. What is lacking at this point is a discussion of the (limited) practical applications of the method, given that: it requires offshore tall-tower data, it was only applied and valid for about 4% of the available data (from Table 3), there are better alternatives (e.g., mesoscale simulations), and the authors are not willing to share their codes.

Reviewer #1 was unsatisfied with the limited changes made to the manuscript to address their concerns. I must agree that the general tone of the manuscript was not changed, the (limited) practical applications of the method were not properly discussed (i.e., only about 4% of the cases from Table 5), and the limitations of the method (only useable where there are tall towers with multi-year measurements) were not addressed. Reviewer #2 did not want to provide a second review, but they had similar concerns. Thus as Editor I am requesting the following minor revisions:

Thank you very much to the reviewers for their comments after the responses and to the associate editor for her reply. We hope that our revision of the paper gives better clarity on our research output and its limitations. We agree it is proof of concept study. Furthermore, we hope our general tone is aligned with the reviewers and editors view on our results.

1. Please modify section 5 title to be "Conclusions and limitations" and add a discussion of the limitations of your method there

The title was changed and an additional paragraph starting in Line 451 was added:

"Although the results are promising, further work is needed to mature this concept of satellite extrapolation with machine learning techniques. This concept is limited by the fixed sampling rate of the satellite observations and the restrictive training area needing multi-year hub-height wind speed observations. This methodology does not have the same practical uses such the alternatives used throughout this work for comparisons (i.e. reanalysis or mesoscale models) but is a step towards improving long-term satellite wind measurements for wind energy purposes."

2. Explain better in the introduction that this is a proof-of-concept paper with limited validation/comparison with other techniques (i.e, ad-hoc mesoscale simulations with or without data assimilation).

This was added in the abstract in hopes to help soften the tone in Line 15 "... results shown in this proof-of-concept study demonstrate the limited applicability ..."

This is further emphasised in the introduction, in Line 70: "As this work is a proof of concept, the model will be assessed in multiple spatial and temporal levels with more established techniques (i.e reanalysis and mesoscale models)."

3. In Section 2, explain why so few data were used at the three FINO sites (4% at most according to Table 3).

We added in Line 165: "Due to the fact that ASCAT collocates with the FINO masts twice a day on average and that all 30 minute input data for the model need to be available for the training process, each mast location only yields a training dataset of under 5000 data points. It should be noted that the choice of 30 minutes averaging of the FINO measurements was to maximize the available collocations. Using larger temporal averaging of one or more hours would represent a larger portion of data with similar wind statistics but would limit the dataset further due missing data in the averaging time window."

Furthermore, we added a point in the discussion about this in Line 328 as "... [the RFM shows] an improvement of over 81 % and 30 %. This result is however limited in that the RFM predictions represent a much smaller fraction of the entire FINO datasets compared to the model outputs due to the data availability of ASCAT defined from the polar orbital paths."

Minor comments:

1. Line 121: unclear still what "daily" means. Is it once a day (if so, at what time?)? Or is it a daily average (if so, over how many hours and what is the temporal frequency of the raw information?)?
Daily as in a daily averaged SST values from various satellite products outlined in the link provided in Line 132. These averaged values are recorded for the start of each day, i.e. each SST grid cell has an associated value at YYYY-MM-DD 00:00:00 UTC. A full description of the dataset is explained in the link provided in Line 132 and the papers cited in Lines 127 and 130.
We added in Line 123 "... foundation temperature available once per day from the Baltic Sea/North Sea ...".
2. Table 5: what are the units of bias? Maybe it is normalized? Then it should be renamed as normalized bias.
It is normalized bias and that is now changed
3. I was confused by Section 3.2: on one hand, it is an application of the same model described in section 3.1, and yet it gives wind speeds at different levels and for the entire 12 year period, thus 5739 concurrent ASCAT observations? Or just 1148? Or all of them?
This part is unclear, I agree. It is the model that was described, trained and evaluated in the previous section, applied to the entire dataset. This is now clarified in the caption as "the random forest model (ML) applied to the entire collocated dataset"

I would also like to ask you to seriously reconsider sharing your codes. Given how little applicability this method already has, if you do not even share the code, then your work will be even less useful. I remind you that "Copernicus Publications encourages authors to also deposit software, algorithms, model code, video supplements, video abstracts, International Geo Sample Numbers, and other underlying material on suitable FAIR-aligned repositories/archives whenever possible." Having said this, it is not required for publication that you share the code, but if you could consider sharing it upon request or similar wording, it would be a wise decision
We have no problem at all with sharing the codes. We have added in the *Data availability* section in Line 461 "The model code is available upon request."